# Effective Usage of Biochar and Microorganisms for the Removal of Heavy Metal Ions and Pesticides

**DOI:** 10.3390/molecules28020719

**Published:** 2023-01-11

**Authors:** Soumya K. Manikandan, Pratyasha Pallavi, Krishan Shetty, Debalina Bhattacharjee, Dimitrios A. Giannakoudakis, Ioannis A. Katsoyiannis, Vaishakh Nair

**Affiliations:** 1Department of Chemical Engineering, National Institute of Technology Karnataka (NITK), Mangalore 575025, India; 2Department of Physics, MVJ College of Engineering, Bangalore 560067, India; 3Laboratory of Chemical and Environmental Technology, Department of Chemistry, Aristotle University of Thessaloniki, 54124 Thessaloniki, Greece

**Keywords:** bioremediation, microbial cell, pollutant, immobilization

## Abstract

The bioremediation of heavy metal ions and pesticides is both cost-effective and environmentally friendly. Microbial remediation is considered superior to conventional abiotic remediation processes, due to its cost-effectiveness, decrement of biological and chemical sludge, selectivity toward specific metal ions, and high removal efficiency in dilute effluents. Immobilization technology using biochar as a carrier is one important approach for advancing microbial remediation. This article provides an overview of biochar-based materials, including their design and production strategies, physicochemical properties, and applications as adsorbents and support for microorganisms. Microorganisms that can cope with the various heavy metal ions and/or pesticides that enter the environment are also outlined in this review. Pesticide and heavy metal bioremediation can be influenced by microbial activity, pollutant bioavailability, and environmental factors, such as pH and temperature. Furthermore, by elucidating the interaction mechanisms, this paper summarizes the microbe-mediated remediation of heavy metals and pesticides. In this review, we also compile and discuss those works focusing on the study of various bioremediation strategies utilizing biochar and microorganisms and how the immobilized bacteria on biochar contribute to the improvement of bioremediation strategies. There is also a summary of the sources and harmful effects of pesticides and heavy metals. Finally, based on the research described above, this study outlines the future scope of this field.

## 1. Introduction

The rapid expansion of industrialization has resulted in the depletion of natural resources and the production of vast volumes of hazardous waste that pollute water and soil, threatening the environment and human health [1]. The deterioration of soil and water quality due to releasing toxic pollutants has become a serious threat around the world. The release of these harmful wastes into the environment occurs in different forms; for example, atmospheric pollutants include noxious gases such as sulfur oxides and nitrogen oxides, while soil and water can be contaminated by organic pollutants (pesticides, hydrocarbons, phenols, etc.) and heavy metals (cadmium, arsenic, lead, chromium, mercury, etc.). Human health can be adversely affected by these environmental pollutants [2] through inhalation or ingestion (Figure 1). Additionally, some pollutants, such as heavy metal ions, can bioaccumulate in the food chain, and these persistent organic pollutants present significant risks to humans and other living creatures.

The accumulation of pesticides and their derivatives is becoming more prevalent, due to the rising population and rapid industrialization. As much as 80% to 90% of pesticides applied to crops in agricultural fields affect non-target life forms; they can relocate or volatilize from the treated area to pollute the air and soil and negatively affect non-target plants. The leaching of these accumulated pesticides leads to the contamination of groundwater and soil [3].

During the last few decades, the separation of pollutants from water systems and soil via several methods has been developed and successfully applied. Recently, technologies such as membrane filtration, ion exchange, and chemical precipitation have been utilized in real-life applications to remove pollutants such as metal ions from polluted areas. Chemical precipitation is a frequently used method for treating heavy metals because it is simple, inexpensive, and effective. However, chemical precipitation results in secondary pollution and eventually leads to additional difficulties in cleaning up the trace contaminants from large areas. Ion-exchange resin offers fast kinetics and is highly efficient for pollutant removal. However, the need for an acidic environment restricts their application in various contexts. Membrane filtration-based technologies can remove toxic substances with high efficiency, but the manufacture of membrane material is usually very complex and at a high cost. Conventional pollutant remediation methods are not eco-friendly and produce toxic chemical sludge. Therefore, there is a serious need to develop efficient and sustainable technologies for remediating toxic environmental pollutants.

Biochar is a carbonaceous material produced through the thermal treatment of different types of biomass, such as crop residues and biosolids [4,5]. Biochar production can be achieved via various processes, including slow or fast pyrolysis, flash carbonization, gasification, hydrothermal carbonization, torrefaction, etc. [6]. The key goal when designing the synthesis of the biochar is that the final material should possess high porosity, a large specific surface area, and elevated surface chemistry heterogeneity, as with oxygen-containing functional groups and minerals [7]. The physicochemical properties of the final obtained biochar can also be tuned by altering the microstructure. Such characteristics encourage biochar’s rising application in (waste) water treatment, soil improvement, and its use in general air, water, and soil remediation [8]. Mechanisms such as physisorption, complexation, precipitation, ion exchange, and electrostatic interaction are involved in the removal of pollutants from aqueous solutions using biochar. Biochar with a high surface area and pore volume exhibits a higher metal–ion philicity because it can be physically entrapped within the pores on its surface [9]. The negatively charged surfaces of biochar can adsorb positively charged metal ions via electrostatic attraction. Compared to other adsorbents/microbial supports, biochar is a low-cost option and a promising candidate for pesticide and heavy metal treatment.

Bioremediation by microorganisms is considered a green technology that is acceptable to the general public. Microorganisms can bioadsorb, bioaccumulate, or biotransform the pollutants permanently at a low operating cost and without the generation of harmful secondary products [10]. Bioremediation can be effective even for contaminants in low concentrations that cannot otherwise be removed by chemical (e.g., incineration) or physical methods. According to some studies, microbial remediation has also been combined with other physical and chemical treatment methods. Hence, bioremediation reduces the health hazards to workers [11]. The microbial degradation of harmful and recalcitrant pesticides is efficient, cost-effective, and eco-friendly, with minimum hazards. The microbial consortia used for remediation have the additional advantage of promoting plant growth in the contaminated site.

Microorganisms can rapidly mutate and evolve in order to withstand environmental stress. The diversity and metabolic activity of the microorganisms are influenced by the presence of heavy metal ions and/or metalloids, which compels the microorganisms to develop resistance systems for overcoming this toxic metal ion stress. Furthermore, microorganisms convert toxic metal ions into inactive forms and can thus be utilized for bioremediation. Pesticide bioremediation involves biodegradation and biotransformation. In biodegradation, biological reactions modify the compound’s chemical structure, decreasing its toxicity.

Microorganism immobilization on biochar is an efficient technology for treating wastewater and soil pollutants [12]. However, less information is available about the degradation of antibiotics, pesticides, heavy metals, PAHs, and other macromolecular organic pollutants that are immobilized on biochar by the microorganisms. Such pollutants are usually remediated via chemical methods or photocatalysis using biochar, which results in the production of free radicals that pose an ecotoxicological risk. As a result, biodegradation is becoming more important, and it is necessary to further investigate unexploited microorganisms for immobilization technology, based on biochar for pollutant biodegradation. However, biochar may cause toxicity to microorganisms according to their particle size [13]. To reduce this toxicity, the appropriate size of biochar must be chosen as a carrier for microorganism immobilization. Therefore, this work will evaluate the properties, influencing factors, strategies of immobilization, and removal efficiency of microbial cell-immobilized biochar (MCB) for the remediation of heavy metals and pesticides. The mechanisms involved in the bioremediation process will be explored. In the published research into the increasing environmental pollution caused by heavy metals and pesticides, much importance is given to remediation techniques. For biological remediation, there are many articles that discuss the role of microorganisms as an effective agent for the remediation of heavy metals and pesticides, as well as the role of biochar as an excellent adsorbent for the above pollutants. However, this review article focuses on the emerging role of biochar as an immobilization support for microbial cells.

## 2. Role of Biochar in the Removal of Metal Ions and Pesticides

### 2.1. Biochar Production, Properties, and Characterization

In general, biochar is a carbon-rich material derived from biomass (such as wood, manure, or leaves) upon thermal treatment at high temperatures in a closed container with minimal or in the absence of air [14]. Various processes, such as pyrolysis, gasification, and hydrothermal carbonization, are applied in biochar generation [15]. Biochar uses include (but are not limited to) carbon capture and storage, capacitive deionization, the Fenton process, microbial fuel cell electrodes, and electrochemical storage [16,17,18].

Biochar has been well established as a low-cost adsorbent that has adsorption capacities similar to carbon-based adsorbents, such as activated carbon, porous graphitic carbon nitride, graphene oxide, etc. The most crucial benefits are: (a) low cost of production, (b) porous structure, (c) simple fabrication on a large scale, (d) eco-friendly nature promoting the cycle of the (bio)economy, (e) multiple surface-functional groups, especially oxygen-containing groups (thus enabling both hydrophobic and polar interactions), (f) ease of modification, etc. [19,20]. In addition, the preference for biochar as a catalyst support for photocatalysis and Fenton/photo-Fenton processes has become prevalent, due to its low cost and high surface area characteristics. The aromatic and other hetero-atom-containing functional groups that are present in biochar also provide moieties that are capable of electron transfer and facilitate the faster and more efficient degradation/reduction of pollutants because of electron delocalization and photo-induced e^−^/h^+^ pairs separation.

#### 2.1.1. Biochar Production

Biochar production usually involves biomass collected from various plant/animal sources or wastewater sludge and thermal treatment using oxygen-deficient conditions, particularly pyrolysis. For instance, plant sources include olive pomace and rapeseed straw cereal waste, whereas animal sources include crustacean shells and animal manure [21,22,23,24,25]. Additionally, municipal wastewater sludge has also been used as biomass for biochar production [26]. The basic composition of biochar predominantly comprises amorphous phases and graphene sheets, as well as various aliphatic cyclic and aromatic groups as a matrix. The temperature of the treatment and the biomass source influence the final physicochemical features. For example, fibrous biomass sources such as wheat/rice straw generate tubular structures [27]. In contrast, the usage of sludge biochar prevents the formation of such structures in the biochar matrix [28].

Pyrolysis in oxygen-free conditions comprises the decomposition of lignocellulosic material, volatile matter release, and the reduction of carbonaceous material for plant biomass [29]. The types of pyrolysis include slow, fast, microwave-assisted, hydro- and co-pyrolysis. Slow pyrolysis operates for hours at lower temperature conditions (300–700 °C), resulting in a higher output percentage of biochar content compared to fast pyrolysis with lower residence time (<2–5 s), higher temperature conditions, and a lower output percentage of biochar. Increasing the temperature can lead to higher carbon content, alkalinity, and elevated specific surface area. In contrast, higher residence time can increase the specific surface area, due to prolonged temperature application.

Variations in high-temperature processes have also been tested in the context of biochar production. Microwave-assisted pyrolysis for biochar generation has also been demonstrated, with variations in absorbable power observed for biochar property analysis, with the demonstrated advantages of larger surface area and improved porosity characteristics [22]. Hydro-pyrolysis is usually conducted within a temperature range of 250–550 °C, with hydrogen gas application, ensuring the hydrocracking of the biomass [19]. Co-pyrolysis involves multiple biomass sources for biochar pyrolysis. The resultant physicochemical properties mainly depend on the biomass sources’ blending ratios and pyrolysis temperature, improving the biochar sample’s pore structure [30]. Gasification is another method of generating biochar in the presence of steam/oxygen at 750–900 °C, with the products being syngas and a low biochar yield. Torrefaction is conducted under oxygen-deficient conditions similar to those for biochar, apart from a temperature of 200–300 °C and a residence time of less than 30 min. Another method explored extensively for biochar production is hydrothermal carbonization, with an operating temperature range from 160 to 800 °C (preferably at lower temperatures) in the presence of water. The low-temperature environment results in higher O/C and H/C content, along with the creation of functional groups on the biochar surface; the process yields a low aromaticity level and low-porosity biochar (hydrochar). The conversion of the non-carbonized (amorphous) part of the biomass into a carbonized form can be enhanced by increasing the pyrolysis temperature, which also increases the aromaticity, π electron availability, etc. [30]. Both the negative effect of pore-size thermal shrinkage due to the collapse of micropore walls and the positive effect of pore-size increment due to the removal of volatile matter can be observed with increasing temperature conditions. Increasing the pyrolysis temperatures also decreases the biochar’s stability in terms of chemical oxidation resistance [31].

#### 2.1.2. Biochar: Physicochemical Properties and Characterization

Several characterization analyses can be conducted to elucidate biochar’s physical and chemical properties. The proximate analysis involves the quantification of ash, fixed carbon, volatile matter, and moisture. High ash and fixed carbon contents are good indicators of high adsorbent capacity. Ultimate analysis, i.e., the quantification of C, H, N, and O composition in biochar samples, especially the H/C ratio and O/C and (O + N)/C ratios, is an indicator of the aromaticity and polarity of biochar [26].

The textural features, with an emphasis on the sizes and the volume of the pores and the specific surface area (S_BET_), are usually estimated via N_2_ sorption tests at 77 K, using the Barrett–Joyner–Halenda (BJH) or density functional theory (DFT) methods for the pore analysis and the Brunauer–Emmett–Teller (BET) theory for the S_BET_. The definition of pore size category (micro-, meso-, and macropores) decides the interaction ability of biochar with the required moiety. For instance, biochar systems with microporous structures would show the lower adsorption capacity of higher molecular-weight pesticides, although a higher one is needed for metal cations [15].

Surface pH analysis, zeta potential, and electrical conductivity can define the range in which biochar–pesticide and biochar–metal ion interactions are maximized. The graphitization and alkalinity of the produced char increase at higher pyrolysis temperatures [32]. Surface functional group analyses, such as cation exchange capacity, Boehm titration, and humic substance analysis, are also used to evaluate the biochar’s adsorption capacity and microbial support. Fourier transform infrared spectroscopic (FTIR) analysis also provides insight into the biochar matrix’s multiple bond formation, with additional information on post-adsorption studies. The solid-state C-nuclear magnetic resonance technique can be used to study the relative abundance of the functional groups and the aliphatic and aromatic hydrocarbon contents [33].

The morphological and structural properties can be explored by scanning electron microscopy (SEM), transmission electron microscopy (TEM), X-ray diffraction (XRD), and atomic force microscopy (AFM). The surface chemistry can be analyzed by IR, Raman spectroscopy, X-ray photoelectron spectroscopy (XPS), potentiometric titration, and Boehm titration. It is always crucial to determine the surface pH and the point of zero charge, since they play a key role in the adsorption performance and activity when biochar is used as an adsorbent in aqueous phases.

The most important techniques for the physicochemical characterization of biochar are presented in Figure 2.

### 2.2. Biochar as Adsorbents

Heavy metals and pesticides can be directly adsorbed onto the biochar’s surface. Modifying the outer surface of biochar via activation by tuning the chemical heterogeneity and/or by anchoring decorating different active species can lead to elevated and selective adsorption efficiency, exceptional stability, easy separation efficiency, and better recyclability [19]. Modification, including physical/thermal activation such as steam (for –OH functional group increment) and CO_2_ activation, ball milling and sonication/ultrasonication, acid treatment (for deashing and demineralization) and base treatment, functional group activation, such as amine-functionalization, impregnation with metal oxides, doping, electrochemical treatment, plasma treatment, etc., can enhance the properties of biochar as an adsorbent [34].

Regarding the analysis of metal ions or pesticide removal using biochar-based materials, Langmuir and Freundlich’s isotherm models are the most established ones. In general, a Langmuir versus Freundlich isotherm comparison explains monolayer adsorption vs. mono/multilayer adsorption, even though this approach is not absolutely correct in the case of studying adsorption in aqueous phases. Other isotherm models/approaches, such as the Jovanovich, Elovich, and Dubinin–Radushkevich (D–R) models, are also used in order to present an additional understanding of the role of adsorption conditions [35,36]. In addition, pseudo-first-order and pseudo-second-order models are the most widely applied models for kinetic studies for biochar–heavy metal/pesticide systems [37].

#### 2.2.1. Removal of Metal Ions

The application of adsorbents for the removal of heavy metal ions involves physical and/or chemical adsorption via electrostatic interactions, ion exchange, complexation, reactions that have taken place on the material’s surface, and/or precipitation [38]. When interacting with biochar, some metal ions undergo reduction and oxidation reactions, precipitation, and co-precipitation [39].

Multiple experimental condition parameters can affect the adsorption and removal capacity. The elevated adsorption of metal ions can be due to an increase in the specific surface area of the biochar as a result of optimizing the synthetic protocol, for instance, modifying the pyrolysis temperature [32]. A high pH directly affects the adsorbent’s surface due to protonation, thus competing with metal ion adsorption [40]. Conversely, in alkaline pH conditions, hydroxy-complex formations can compete with other ions and impede adsorption [21]. Preferably, the point of zero-charge pH should be in the acidic region to efficiently adsorb metal ions and form complexes with a negative surface charge [41]. Cation-exchanging capacity also plays a crucial role in metal ion adsorption. For instance, Ma et al. [42] discovered that cation exchange significantly contributed to removing Cu^2+^ from lobster-shell-derived (HCl-treated) biochar, with 53–74% removal contributed by the cation exchange.

Biochar surface modifications are primarily conducted to improve the adsorption efficiency, and some of them are summarized in Table 1. A zirconium and iron composite with sludge biochar was generated to increase As^5+^ adsorption via complexation. The Zr-Fe biochar composite had a maximum adsorption capacity of 62.5 mg/g, compared to the pristine biochar capacity of 15.2 mg/g. The probable mechanism was suggested as the inner-sphere complexation of As^5+^ on the Zr-O-Fe surface [36]. Khan et al. [43] studied MoS_2_-modified magnetic biochar with a maximum adsorption capacity of 139 mg/g, and hypothesized the presence of complexation, cation exchange, and Cd-π interactions. The deashing of biochar with acid solutions and potassium acetate improved lead adsorption, due to the pore size increment (unblocking SiO_2_ particles out of biochar) and complexation of Pb^2+^ and C=C (π-electrons) [40].

#### 2.2.2. Adsorption/Removal of Pesticides

Studies indicate that increased pesticide concentration and adsorption time has an asymptotic effect on adsorption capacity, whereas the adsorption capacity is enhanced by the increases in biochar concentration. The common mechanisms for pesticide adsorption onto biochar are the hydrophobic effect, π–π electron donor–acceptor interaction, pore filling, electrostatic interactions, ionic bonding, and H-bonding [26,52].

Several studies regarding the adsorption/removal of pesticides by biochar have been evaluated in Table 2, wherein the parameters of biochar pyrolysis temperature and surface modifications have been compiled, along with the adsorption capacity values. The pyrolysis temperature has a similar effect on biochar-pesticide adsorption as on biochar-heavy metal adsorption. The adsorption of carbendazim on dewatered sludge biochar was at a maximum at 700 °C, owing to the increased surface area and the increment in the partition coefficient [26]. Pore size governs the definitive adsorption capacity for pesticide–biochar interaction. Dichlorvos and pymetrozine had molecular sizes that were comparable to pore diameter; thus, adsorption was facile in both cases [53]. A decrease in the original biochar’s H/C and O/C atomic ratios is expected to enhance the π–π electron donor-acceptor interactions, contributing to the sorption of certain pesticides, such as oxytetracycline and carbaryl [24]. Binh and Nguyen [52] concluded that a pH of 2 is a more favorable condition for the adsorption of 2,4 dichlorophenoxy acetic acid on corn-cob biochar based on the electrostatic interactions. In addition to the inherent functional groups and mechanisms involved in metolachlor adsorption onto biochar, Liu et al. [54] incorporated fulvic acid and citric acid into walnut-shell biochar that augmented the functional groups with oxygen, as shown in Figure 3. The removal capacity was also observed to decrease after 3 cycles in the metolachlor-simulated sewage biochar system.

### 2.3. Biochar as a Bioremediation Catalyst Support

The physicochemical properties of biochar that enable it to be an effective catalyst support include its large surface area, multi-scale porous structure, and surface functional group. Chen et al. [57] studied the volatilization of Hg^2+^ using the *Pseudomonas* strain, DC-B1, with biochar. The combined application of biochar and microbial strain resulted in the greatest Hg removal. Qiao et al. [58] demonstrated the stimulation of the microbial reduction of As^5+^ and Fe^3+^, using oil palm fiber-derived biochar in synergy with soil microbes extracted from paddy for both studies. Biochar amended microcosm possessed a higher As^5+^ concentration than the control, indicating that biochar had an affinity to As^5+^ and Fe^3+^. Both moieties were reduced in the biochar-amended microcosms since microbes drove the reduction reactions, and biochar behaved similarly to an electron shuttle. Qiao et al. [59] summarized the As^5+^ reduction with biochar and lactate. This reduction resulted in the identification of three ways: (i) Fe^3+^ reduction by microbial cells facilitated As^5+^ release; (ii) expression of As^5+^-respiring gene transcripts in dissimilatory As^5+^-reducing bacteria; (iii) the functioning electron transfer between the metal and As^5+^-reducing bacteria.

Biochar is often employed as a good carrier in improving the photocatalytic activity of metal oxides. As a stable and inexpensive carbonaceous material, biochar effectively reduces the recombination rate of photogenerated electron-hole pairs, due to its excellent conductive property. An et al. [60] developed biochar-supported α-Fe_2_O_3_/MgO composites for photocatalytic degradation of organophosphorus pesticides and obtained a degradation efficiency of 90% in 80 min. Huang et al. [61] utilized pristine and manganese ferrite-modified biochar for Cu removal, confirming the role of biochar being principally an oxide carrier instead of an adsorbent. In addition, a preference for biochar as a carrier for photocatalysis and Fenton/photo-Fenton processes has been prevalent, due to its low cost and high surface area characteristics. The utilization of lignin-biochar as a catalyst support for LaFeO_3_ in the catalytic photo-Fenton process had a positive effect on the degradation efficiency of pollutants, owing to enhanced adsorption capacity, a reduction in the charge transport resistance between LaFeO_3_ and lignin-biochar, and the presence of oxygen-containing functional groups [62].

Several studies have been conducted for enzyme-immobilized biochar, particularly with laccase utilized as an enzyme to degrade pollutants [33]. The basic biochar–enzyme immobilization techniques are adsorption and covalent bonding. Comparatively, fewer instances of enzyme–biochar systems for the degradation of pesticides have been studied. Wang et al. [63] used laccase-immobilized biochar to degrade 2,4-dichlorophenol and obtained 64.6% degradation. The immobilized laccase improved the cation exchange capacity, organic matter content, stability, and catalytic degradation effect. A general outline for adsorption and the removal mechanisms for metals and pesticides via biochar systems are shown in Figure 4.

## 3. Role of Microorganisms in the Removal of Metal Ions and Pesticides

The surface area of microorganisms exhibits higher biological activity relative to their volume, resulting in greater interaction with their immediate environment. Thus, they can adapt and survive in polluted areas with the subsequent removal or detoxification of the pollutant [64]. The microorganisms use different strategies for their survival, including surface adsorption, micro-precipitation, extracellular or intracellular sequestration, reduction, enzymatic degradation, etc. Bioremediation is possible only when microbial activity and growth are allowed by environmental conditions. In certain situations, environmental factors can be altered to allow microbial population growth to eliminate contaminants [11]. As shown in Figure 5, various factors influence microbial degradation:(i)Environmental factors

The pH can affect bioremediation by changing metal bioavailability; for instance, a decrease in soil pH value generally causes an increase in metal bioavailability [65]. This is because, at lower pH, the exchangeable capacity between metal cations and H^+^ on the surface of soil particles is more prominent than at higher pH. Additionally, an optimum pH is essential for microbial growth, and some microbial degradation processes can be inhibited at an extreme pH. Temperature is another crucial factor influencing the bioremediation of metals and pesticides [65]. The solubility of these contaminants is increased at higher temperatures, which leads to their increased bioavailability. The physical nature and chemical composition of several organic pollutants and their adsorption-desorption mechanism are governed by temperature. Temperature also influences microbial growth, activity, and degradation potential. Furthermore, the soil moisture content is another parameter that affects the bioremediation process. A low soil moisture content limits the growth and metabolism of microorganisms, while high values can reduce soil aeration.

(ii)Type of microorganism and degradation capacity

The microorganism that is selected for biodegradation should be able to survive in a high-contamination environment and should be evaluated first for its degradation capacity before employing it for in situ remediation. The survival of these strains can be ensured by providing favorable growth conditions. It is also important to note that microbial strains selected for pollutant removal may need to meet certain ecological requirements. One such requirement is that the strains should be non-pathogenic. For instance, *Staphylococcus aureus*, as a typical pathogen, was resistant to many antibiotics and showed high bioremediation efficiency for heavy metals such as Cr and U through bioprecipitation [66]. However, certain metabolites that formed during the degradation of contaminants can be toxic. Therefore, deeper investigations of ecological security and the metabolic functions of microbial cells are indispensable before their possible application in environmental pollution control.

(iii)Bioavailability of the contaminants

The bioavailability of the contaminants can be defined as the fraction of a contaminant in a specific environment that is either adsorbed or degraded by the microbial cells within a given time. The control of bioavailability is dependent on the diffusion, uptake, and desorption of the contaminants. The slow mass transfer of contaminants into degrading microbes reduces their bioavailability. The significance of bioavailability depends very much on the properties of the pollutant, microorganism, and characteristics of the contaminated site [11].

(iv)Aerobic or anaerobic operating conditions

Depending on the type of organism and contaminant, bioremediation can be either aerobic or anaerobic. Most bioremediation systems work under aerobic conditions, but to effectively degrade the recalcitrant molecules, it is better to run the microbial degradation tests under anaerobic conditions. Apart from the abovementioned factors, the properties of the contaminated site (soil type, soil porosity, soil nutrients) and the properties of the contaminants (structure, hydrophobicity, recalcitrance, toxicity, solubility, and leaching ability) are also important in bioremediation.

### 3.1. Removal of Heavy Metals Using Microorganisms

The removal of heavy metal ions by microorganisms is considered economical and sustainable. Any environmental stress can be withstood by microorganisms through rapid mutation and evolution, leading to toxic heavy metal resistance. They can sequester heavy metal ions, either intracellularly or extracellularly. Additionally, microorganisms can transform and reduce the metal ions to inactive forms. Table 3 summarizes the microorganisms used for various metal ion remediation conditions in recent years.

The factors influencing heavy metal remediation by microbes generally include pH, temperature, biomass concentration, the presence of other pollutants, etc. The inherent pH of the system defines the charges of the surface functional groups present on the microbial surfaces; pH in an unsuitable range may affect microbial growth. This shows pH to be an essential parameter in the degradation and removal of heavy metals by live biomass [81]. The pH also has an effect on the solubility of metal ions in the microbe-heavy metal system. A decrease in soil pH leads to an increase in the bioavailability of metals, thereby resulting in higher biosorption efficiency, as studied by Zhang et al. [73]. Another essential parameter in microbial growth and proliferation is the system’s ambient temperature. With an increase in temperature, the solubility of metal ions increases; thus, the bioavailability of metals also increases [81]. High biomass or sorbent concentration will increase the overall biosorption efficiency, but any interference between binding sites reduces the specific metal ion uptake. The removal or adsorption of a particular heavy metal by microorganisms can also be positively or negatively affected by the co-existence of other metal ions.

#### The Mechanism of Heavy Metal Removal by Microorganisms

Microorganisms can adopt several mechanisms in order to survive in heavy-metal toxicity conditions. These mechanisms are depicted in Figure 6 and include biotransformation, extracellular polymeric substances secretion, metallothionein synthesis, etc. Heavy metal degradation by microorganisms can be described in two ways: biosorption and bioaccumulation.

Biosorption is the reversible physicochemical interaction of living (or dead) biomass or biomass-secreted products that act as biosorbents with sorbate molecules (e.g., metal ions). It was previously categorized as metabolism-dependent and metabolism-independent biosorption. Recently, the former has been widely accepted as bioaccumulation (also called active biosorption), and only the metabolism-independent processes are considered to be biosorption [82]. A metabolism-independent mechanism occurs passively on the dead or the living biomass cell surface. However, the biosorption of metal ions carried out by dead biomass is superior to that carried out by living cells. Cheng et al. [78] studied the biosorption of Cd^2+^ in the living and dead cells of the microalgae *Chlorella vulgaris*. The dead algal biomass removed 96.8% of the total cadmium, while the live algal biomass achieved 95.2% of cadmium adsorption. The steps involved in toxic heavy metal biosorption include binding the metal ions to various extracellular functional groups present on the microbial cell wall, via surface precipitation, chemical bonding (complexation/chelation), adsorption, or ion exchange. Physical adsorption depends on intermolecular or inter-ionic attraction forces. Complexation or chelation occurs due to the dative covalent bonds between metal ions, surface functional groups, and the ligands of biomass. When metal ion concentrations are higher than the solubility limit, surface precipitation or micro-precipitation has been observed. The exchange involves electrostatic interaction between the metal cations and the negatively charged functional groups on the cell surface; the interchange of the cations resulted in the metal ion being bound to the surface [82,83]. Surface-binding is found to be the principal phenomenon governing the biosorption of metal ions [84]. Physical modifications have been suggested to provide a cumulative effect on the biosorption capacity of the microorganisms by removing surface impurities or through the production of metal-binding sites. Li et al. [74] investigated the biosorption ability of a lactic acid bacterium, *Weissella viridescens* ZY-6, for Cd^2+^ removal from the aqueous solution, and achieved a 69.45–79.91% removal of Cd^2+^ from three kinds of juices: tomato, apple, and pear juices.

The extracellular sequestration of metal ions often occurs due to various biological structures produced by microbial cells, including extra-cellular polymeric substances, siderophores, glutathione, and biosurfactants. Under heavy metal stress, microorganisms often secrete extra-cellular polymeric substances or exopolysaccharides (EPS) as a protective response. EPS are constituted of proteins, lipids, complex carbohydrates, nucleic acids, uronic acid, humic acid, etc., which prevent the entrance of heavy metals into the cell [68,85]. Generally, EPS contain negatively charged functional groups and can interact electrostatically with heavy metals, resulting in the immobilization of the metal ions within the EPS. Some examples include the accumulation of Pb^2+^ and Zn^2+^ in the soluble EPS secreted by *Oceanobacillus profundus* KBZ 3-2 [68], and Pb^2+^ adsorption onto EPS of *Enterobacter* sp. FM-1 [69] and Cd^2+^ adsorption onto the EPS secreted by a living cyanobacteria, *Synechocystis* sp. PCC6803 [86]. Siderophores are secreted by microbes and act as metal chelators, with an extreme affinity for ferric iron. They can reduce the metal’s bioavailability and toxicity by binding metal ions with variable affinities that have a similar chemistry to that of iron [87].

Biosurfactants are amphiphilic compounds that are produced extracellularly by microorganisms for the solubilization, desorption, complexation, and mobilization of pollutants in solutions. The induction of biosurfactants in microbe-heavy metal systems facilitates the extracellular sequestration and formation of biosurfactant–metal complex [88]. Rhamnolipids produced by *Pseudomonas aeruginosa* showed 53% As, 90% Cd, and 80% Zn extraction capacity from contaminated soil [89]. Ayangbenro and Babalola [90] observed that a lipopeptide biosurfactant generated by *Bacillus cereus* NWUABO1 could remove 69% of Pb, 54% of Cd, and 43% of Cr from the soil. Several microorganisms, including *Pseudomonas* sp., *Bacillus subtilis*, *Candida tropicalis*, *Candida* sp., *Burkholderia* sp., and *Citrobacter freundii* can produce biosurfactants, demonstrating heavy metal removal capacity [88].

Biosorption has been determined to be simple, fast, reversible, and inexpensive compared to bioaccumulation and can concentrate heavy metals, even from a very dilute aqueous solution. The advantageous properties of biosorption include the presence of multi-functional groups and the uniform distribution of binding sites on the cell surface, low operational cost, the absence of metal toxicity limitations, minimal preparatory steps, high efficiency and selectivity for metal ions, no production of secondary waste, and the possibility of the toxic heavy metal recovery and reusability of the biosorbent [84,91]. Several microbial strains have been identified to show multi-metal resistance and remediation abilities. Nokman et al. [92] isolated a *Pseudomonas putida* strain from effluent water generated from a tannery that exhibited resistance to Ag^2+^ and Co^2+^ and enhanced resistance to lead and chromium. Conversely, bioaccumulation is the metabolism-dependent active transportation of metal ions across the membrane into the living cell, as represented in Figure 6. The microorganisms selected for bioaccumulation should have specific properties, such as adaptation to the polluted environment, resistance to high loads of metal ions, and a mechanism of intracellular binding [93]. The mechanism consists of two steps; the first step is identical to biosorption and involves the attachment of heavy metals to charged functional groups on the cell surface. The second step is metabolism-dependent, relatively slow, and involves the penetration/transport of a metal-ligand complex into the cell membrane. The subsequent interaction of the complexes with intracellular metal-binding proteins (such as metallothionein and phytochelatins) occurs within the cell, leading to bioaccumulation [85]. Metallothioneins (MTs) help to regulate the intracellular metabolism of metals and protect against oxidative stress and toxic heavy metals [86,87]. Engineered recombinant *E. coli* expressed the *Corynebacterium glutamicum* metallothionein gene and achieved improved intracellular biosorption of Pb^2+^ and Zn^2+^. Hu et al. [86] constructed a bio composite of immobilizing metallothionein, expressing *Pseudomonas putida* for the sorption of Cu^2+^. Similarly, phytochelatins are metal-binding proteins that are analogous to the metallothioneins produced from microalgae, which can also chelate and detoxify heavy metal ions intracellularly.

### 3.2. Removal of Pesticides Using Microorganisms

The major types of pesticides and persistent organic pollutants include insecticides, herbicides, and fungicides. As with heavy metals, the microbial remediation of these persistent pesticides is economical and sustainable, compared to physical or chemical removal processes. It involves the degradation of complex pesticide molecules into simpler inorganic chemicals. Table 4 includes the commonly used microorganisms for the removal of pesticides. Indigenous soil microbial consortia have been more effective for the microbial degradation of pesticides than the non-indigenous strains, as non-indigenous strains are exposed to pesticide-contaminated regions exhibiting unfamiliar conditions. Several studies have reported on the degrading ability of indigenous microbes. Some of them show organophosphate degradation by indigenous *Kosakinia oryzae* [94], herbicide glyphosate degradation by *Providencia rettgeri* [95], and herbicide atrazine remediation by indigenous microbial consortia [96]. Individual or mixed microbial cultures can degrade the various sources of pesticides. Single microbial cells abide by their metabolic pathways for pesticide degradation, whereas mixed microbial cultures can achieve the same result through coupled metabolic pathways [97]. Thus, pesticides can rapidly be degraded by applying the combined microbial consortia isolated from indigenous sites.

However, certain recalcitrant pesticides have resilience against biodegradation by the indigenous microbial community. In such situations, bio-augmentation and bio-stimulation are considered promising approaches for the remediation of contaminated sites. Bio-augmentation involves the introduction of specific exogenous microbes to improve the degradative capacity of the contaminated sites. The two main strategies of bio-augmentation are autochthonous bio-augmentation, where the microbes are isolated from the same site and then re-injected, and allochthonous bio-augmentation, where the microbes are cultured from another site [107]. In one study, bio-augmentation with *Paenarthrobacter* sp. W11 significantly accelerated the degradation rate of atrazine in soil and dampened its toxic effect on wheat growth [108]. The success of bio-augmentation strategies depends on several factors, including the selection of appropriate microorganisms, the target pollutant’s bioavailability, and the inoculum’s survival capability in the toxic environment [11,109].

Bio-stimulation can be performed by providing the necessary nutrients or electron acceptors, such as oxygen or nitrate, to promote the proliferation of indigenous microbes. Aldas-Vargas et al. [110] investigated the biodegradation of herbicides, namely, mecoprop-p and 2,4-dichlorophenoxyacetic acid (2,4-D), in groundwater. They concluded that bio-stimulation with oxygen and dissolved organic carbon had the potential for field application. Raimondo et al. [111] bio-augmented lindane-contaminated soil with actinobacteria (mixed culture) and bio-stimulated it with sugarcane filter cake, further noticing enhanced lindane removal, along with microbial cell counts and enzyme activities.

The removal of pesticides depends, firstly, on the optimal conditions of the biomass, its survival and activity, and, secondly, on the pesticide’s chemical structure, along with several biotic and abiotic factors, such as suitable microbial strains, nutrient availability, salinity, pH, temperature, etc. [112]. In the case of the in situ remediation of soil contaminated by the extensive use or overuse of pesticides for agricultural purposes, the growth of pesticide-degrading soil microbes depends on the soil characteristics [11].

#### Mechanisms Involved in Pesticide Removal by Microorganisms

There are several mechanisms by which microorganisms transform pesticides into their non-toxic forms in a contaminated site. Some include the surface adsorption, enzymatic degradation, or co-metabolism of the pesticide molecules, as depicted in Figure 7. Adsorption of the pesticide molecules is categorized as a passive process and involves the direct interaction of molecules with the microbial cell surface. As a result, the efficiency of pesticide adsorption by microorganisms is primarily determined by the available surface-active groups. The ultimate result of adsorption is the reduced mobility of the toxic pesticides. The extent of removal and the degradation efficiency are influenced by various components, such as the charge, polarity, solubility, volatility, and solubility of the pesticide molecules. Extra-cellular polymeric substances (EPS) and biosurfactants produced by the microorganisms also aid in the removal of pesticides. EPS can be produced by the microbial cell as a byproduct of pesticide degradation. This approach can have two benefits: (i) the reduction of excess toxic pesticides, and (ii) the production of EPS, which can have further environmental applications. Gupta et al. [113] observed 98% carbofuran degradation within 96 h by *Cupriavidus* sp. with simultaneous EPS production. Satapute and Jogaiah [114] reported that surfactin, a biosurfactant produced by a bacterial strain, could degrade 91% of difenoconazole.

Microbial enzymes can catalyze the breakdown of pesticides. The enzymatic degradation processes may include an alteration in the structural components, the removal of undesirable pesticide properties, oxidation, and reduction [115]. Dash and Osborne [116] investigated monocrotophos degradation by *Bacillus aryabhattai* (VITNNDJ5) instead of the bacterial enzyme. The enzymatic degradation of pesticides can either be performed by intracellular enzymes that are present in the microbial cell or by extracting the enzymes capable of degradation from the cells. Sirajuddin et al. [100] isolated the *E. coli* IES-02 strain from a site contaminated with the organophosphate malathion, and the strain showed efficient degradation, utilizing it as the sole carbon source. They also purified carboxylesterase enzyme from the IES-02 strain and achieved 81% malathion degradation under optimized conditions within 20 min, whereas the IES-02 cell degradation was completed from 99.0% to 95.0% within 4 h. However, the extracted enzymes can be affected by solution properties, such as pH, temperature, etc. Depending on the environmental factors, enzymes may lose their degradation potential due to varied ambient conditions [117]. Oxidation, hydrolysis, alkylation, and dealkylation reactions have been predominantly observed in the microbial degradation process [118]. Some studies that have reported enzymatic degradation are on cypermethrin by esterase and laccase [119], carbendazim by carbendazim hydrolase [120], malathion by phosphotriesterase [121], and isoproturon, procymidone, chlorpyrifos, dichlorophos, and monocrotophos by laccase [122,123,124]. The enzymatic biodegradation mechanism of pesticides is often complex, and this diverse biodegradation pathway needs further investigation to understand enzyme involvement properly.

### 3.3. Challenges of Using Microorganisms as a Catalyst

The microbial degradation of metal ions and pesticides tends to be an appealing approach for bioremediation, even though certain challenges hinder their commercial application. These include: (i) the loss of microorganisms or reduced microbial survival because of the toxicity to microorganisms at a higher metal ion or pesticide concentration, (ii) reduced microbial proliferation, (iii) uneven microbial growth with high concentrations of the pollutant, (iv) the washing out of the microbial cells during the application, (v) the longer time required for the completion of the process, (vi) the presence of other co-existing metal ions and organics that can positively or negatively affect the remediation process.

Microbial immobilization on a support material can overcome the above drawbacks by fixing the free microbial cells to a specific carrier, either chemically or physically, and keeping them active for longer. An ideal carrier provides operational stability and cell protection from the toxic external environment, leading to efficient biodegradation. A support material retains the microbes and contributes to the sorption of the pollutants [125]. Hence, immobilizing the microorganism accelerates the pollutant’s biodegradation capacity, enhances the robustness of the immobilized strains, and improves their tolerance to high pollutant concentrations.

## 4. Microbial Cell-Immobilized Biochar for the Removal of Metal Ions and Pesticides

Bioremediation with free microbial cells is generally inefficient, due to the lesser amount of microbes utilized for degradation, microbial loss, and the inhibition of growth and functioning from indigenous microorganisms [126]. Immobilizing the microorganisms creates a safe environment for microbial cells to perform specific functions, such as highly efficient physiochemical sorption and microbial metabolism. Pollutant adsorption/binding on the carrier material allows the degrading cells to outcompete indigenous microbes, overcoming the limitations of using free cells for bioremediation [127]. Biochar has been a prominent carrier for microbial cell immobilization, due to its minimal toxicity and abundant generation. Immobilized microbes have commonly been observed for better remediation efficiency than pristine biochar or free cell [128].

### 4.1. Immobilization Methods

Biochar-immobilized microorganisms are produced through the adsorption of microbes on biochar, entrapment with the help of crosslinking materials, or a combination of both methods. Adsorption is a simple and inexpensive method for immobilizing microorganisms [129,130]. Adsorbed cells colonize the biochar after being transferred from a bulk solution to its surface. The adsorption technique involves physical interactions, such as van der Waals forces, ionic interactions, and hydrogen bonding between the surface functional groups of microorganisms and functional groups on the surface of carriers, particularly the oxygen-containing groups, such as carboxylic, phenolic, and sulphonate groups. Microorganisms have a low affinity for carriers; there will thus be a high rate of desorption of cells from carriers [125]. As a result, appropriate carriers with high cell-binding characteristics are required for improved remediation. With a relatively weak interaction between the carrier and microbial cells, immobilization does not affect the intrinsic structure of the original microbes if the adsorption method is utilized. As a result, this method is better suited for immobilizing viable cells and biodegrading pollutants in the laboratory. Entrapment is a standard method of physical immobilization that is irreversible and provides better stability of microbes than adsorption [126]. Due to the improved stability of the thus-prepared immobilized cells, the entrapment method is preferred and is exercised in industrial applications for pollution abatement.

### 4.2. Factors that Influence Bioremediation Using Immobilized Microorganisms

The effective pollutant removal capacity of MCB is affected by pollutant concentration and its bioavailability, the incubation time of the cell, and various parameters, such as temperature, pH, etc. The biochar-immobilized microorganism technology requires a thorough understanding of the best conditions for maximum contamination removal.

Initial pollutant concentration influences the removal of pollutants, wherein setting the initial pollutant concentration until the saturation point increases the adsorption capacity of the biosorbed pollutants per unit weight of MCB [131]. The bioavailability of pollutants is defined as the total amount of a contaminant that is either available or that may be made available for uptake by microorganisms from its surroundings within a given period. The significance of bioavailability depends on the pollutant’s physicochemical properties, microorganisms, and contaminated site characteristics [11]. Incubation time is another critical parameter affecting bioremediation because it has been observed to affect the growth pattern of microorganisms directly. *Proteus mirabilis* YC801, immobilized on biochar, achieved a 42.5% Cr bioreduction and adsorption capacity after 6 h of incubation [132]. The temporal requirement is high for microbial degradation, and the reaction time for complete degradation is higher than that for other removal processes. The time scale of the microbial degradation process can be reduced by selecting suitable microorganisms with quicker growth phases for pollutant degradation or removal. However, choosing biochar with a high adsorption potential for pollutants is critical for reducing the bacterial adaptation time.

The pH value also influences microbial metabolic processes, particularly growth, cell membrane transport, the zeta potential of sorbate, and changes in the sorbent surface characteristics [133]. Huang et al. [132] observed an increase in Cr^6+^ reduction with a pH increment from 6.0 to 7.0, showing a maximum removal of 83.7% at pH 7.0. However, alkalifying the Cr^6+^-MCB system from pH = 8.0 to pH = 10.0 inhibited the removal capacity of MCB for Cr^6+^. Similarly, the highest Cr bioreduction was found at 30 ℃, similar to the optimal culture temperature for the strain. Bioreduction significantly decreased with a further increase in temperature above 30 °C, which might be attributed to the loss of cell viability and the inhibition of the essential enzymes and proteins responsible for microbial growth and biodegradation at elevated temperatures [12,132].

Similarly, temperature and pH significantly influenced tebuconazole degradation by *Alcaligenes faecalis* WZ2, and degradation efficiency was strongly correlated with bacterial growth [125]. Tebuconazole degradation efficiency reached 88.5% under ideal conditions (a temperature of 30–35 °C and a pH of 6–8). Because of bacterial growth inhibition and a decrease in the catalytic activities of microbial enzymes involved in tebuconazole degradation, the efficiency was significantly reduced below the ideal temperature and pH.

### 4.3. Heavy Metal Ions and Pesticide Removal Using MCB

The advantages of immobilizing cell systems onto carriers in the bioremediation of metal ions and pesticides are far superior to those of using biochar or free cells alone [134]. Pollutant transfer into the microbial community from the contaminated sites can be enhanced by immobilizing the microbial strains onto biochar. Biochar can enhance the biological community composition of the soil through physisorption; in return, these microorganisms, adsorbed on the biochar surface, have a metabolizing capability for the pollutants present in the soil [135]. The porous structure of biochar enhances the growth and reproduction of the microorganisms and can also act as a source of nutrients for the microorganisms [136]. The immobilization also ensures that the microorganisms are assimilated for degradation to form biofilms around the porous structure complex of the biochar microbes [136]. Biochar can alleviate the contaminant concentration and reduce the inhibitory effect of these contaminants on the growth of microorganisms via the adsorption and subsequent decrease in contaminant concentration in soil/aqueous medium [137].

Using biochar and bioremediation in tandem with functional microbial strains is a viable and emerging strategy for the long-term remediation of contaminated water and soil. Numerous microbial strains with strong metal tolerance or adsorption capability have been isolated and used for bioremediation, either as free-living cells or by immobilizing a microbial cell with a specific carrier substance. Metal-tolerant microorganisms immobilized on biochar have been used as a bio-augmentation method to improve heavy metal phytoremediation, indirectly reducing heavy metal contamination in soil. Incorporating bacteria immobilized on biochar into the soil may indirectly improve Cd removal by promoting plant growth and the phytoremediation effect [138]. Cd-resistant bacteria immobilized on biochar improved the phytoextraction efficiency by *Chlorophytum laxum* R. Br. via cadmium phytoaccumulation in the shoots and roots, and Cd translocation from the roots to the shoots. Insoluble phosphate solubilization can be achieved via microbial phosphate solubilizers (PSB). Teng et al. [134] observed that combining PSB and biochar improved Pb^2+^ immobilization by forming a stable crystal texture on its surface. Zhang et al. [139] used the PSB bacteria *Pseudomonas chlororaphis* for lead removal. However, the organism could not proliferate in indigenous bacteria, whereas the addition of PSB-immobilized biochar (PIB) improved bacterial growth and reduced Pb concentrations to less than 1 mg/kg. As a result, soil inoculation with PIB can be used as a substitute for Pb immobilization, avoiding the secondary pollution caused by phosphorus toxicity.

The microorganism immobilization with biochar carrier was also influential in remediating soil polluted with a combination of heavy metals. Tu et al. [140] introduced *Pseudomonas* sp. NT-2, loaded onto maize straw biochar, into Cd-Cu mixed soil. The application of *Pseudomonas* sp. NT-2-loaded biochar effectively reduced the bioavailability of Cd and Cu and increased the soil enzymatic activities in the soil system. Qi et al. [135] used three strains of mixed bacteria, *Bacillus subtilis*, *Bacillus cereus*, and *Citrobacter* sp.-loaded biochar for U and Cd removal. They discovered that MCB promoted growth in celery and reduced the U and Cd phytoaccumulation, compared to free cell and biochar treatments. Research on Cr^6+^ removal by immobilized microorganisms with biochar has attracted increased interest recently. The metal ion-resistant bacterium, *Proteus mirabilis* YC80, was immobilized using biochar derived from the bloom-forming cyanobacterium, *D. flos-aquae* [132]. The ability of biochar-immobilized *Proteus mirabilis* PC801 to remove Cr^6+^ was superior, compared to a free cell. The removal efficiency of Cr^6+^ by PC801-immobilized biochar was 100%, with 87.7% total Cr immobilized on the carrier and only 12.3% Cr^3+^ remaining in the solution. Table 5 includes microbial cell immobilized biochar reported for heavy metal and pesticide abatement.

Physical adsorption, ion exchange, surface complexation, precipitation, and biotransformation are some of the mechanisms involved in MCB-mediated heavy metals removal (Figure 7). Biochar containing oxygen functional groups, mineral components such as carbonates and phosphates, and microbial surface functional groups contribute to the removal of metal cations. Shen et al. [143] investigated the mechanism of cadmium removal using biochar-immobilized microalgae. They discovered that electrostatic attraction, surface complexation, and ion exchange were responsible for cadmium removal (maximum adsorption 217.41 mg/g) from wastewater. Similarly, Tu et al. [140] noted that surface complexation with different functional groups on cells, cation exchange, and surface complexation on biochar contributed to the enhanced stability of Cd^2+^ and Cu^2+^ in the contaminated soil. Microorganisms secrete enzymes that mediate redox reactions and surface complexation. These are the mechanisms involved in removing As^3+^, As^5+^, Cr^6+^, U^6+^, and Mn^2+^. Youngwilai et al. [147] examined the mechanism of Mn^2+^ removal by the *Streptomyces violarus* strain, immobilized on biochar. They found that the two processes, namely, biological oxidation by the immobilized strain and adsorption by biochar, work together.

The presence of multiple contaminants at a particular contaminated site is a widespread phenomenon that could severely affect the microorganisms’ remediation potential [148]. This limitation can be addressed by the associative effect of the benefits of biochar and the microorganisms via the immobilization of functional bacteria (such as organic contaminants–degraders) on biochar, as this could potentially remediate various types of contaminants. The application of biochar-microbial complex also increased the soil microbial and enzymatic activity, along with conducting the simultaneous bioremediation of multiple contaminants in several studies [126,148]. Several studies report that the degradation efficiency by biochar-immobilized bacterial consortia in co-contaminated sites is significantly enhanced compared to the free bacteria, due to their bioaugmentation abilities. For instance, Li et al. [149] immobilized PAH-degrading bacteria (*Citrobacter* sp.) into biochar, increased the degradation rate of PAH and reduced the toxicity of Ni by bio-transforming the available Ni into a stable form.

Pesticide degradation can be enhanced by introducing exogenous free cells to polluted soil. However, this method has several drawbacks, including the growth and survival of microbial cells, inadequate nutrients, lesser adaptability to surroundings, and competition with native microorganisms [139,150]. Immobilizing the exogenous pollutant-degrading bacteria on a support material can be an alternative strategy. This can be an ideal environment for their survival in different soil conditions [151]. Microorganisms immobilized in biochar have the potential to directly or indirectly reduce environmental pollution, while also allowing for the long-term maintenance of catalytic activity. Due to its superior porosity, ample surface area, and functional groups, biochar is an ideal medium and a rich nutrient composition for immobilizing and reproducing microbial cells [125].

Biochar can improve the soil’s pollutant adsorption capacity while providing the nutrients for microbial growth and function [152]. Adsorption and covalent-binding methods were used to immobilize *Pseudomonas putida* onto coconut fiber-derived biochar. The efficacy of MCB in paraquat removal from contaminated water was studied by Ha et al. [129]. After 48 h of incubation, MCB could convert paraquat to 4,4-bipyridyl and malic acid. According to Wahla et al. [148], the immobilization of the MB3R consortium was achieved on biochar-remediated soil contaminated with metribuzin. The immobilization of a microbial consortium on biochar increased the rate of cypermethrin degradation and removal efficiency while lowering the cypermethrin’s bioavailability to indigenous organisms [153]. Sun et al. [125] isolated and identified *Alcaligenes faecalis* WZ-2 as a tebuconazole-degrading strain and supported it on wheat straw biochar as a carrier. The biochar-immobilized WZ-2 reduced the half-life of tebuconazole in soil from 40.8 to 13.3 days and affected the microbial population and enzyme activities in polluted soil.

## 5. Conclusions and Future Prospective

Recent research on removing heavy metal ions and/or pesticides using biochar and microorganisms has revealed their enormous potential. Biomass-derived materials, such as biochar, have gradually been established as a viable platform for advancing the design and development of carbon-based materials and their suitability for various uses, such as, for instance, environmental remediation applications. A plethora of biochar production and activation approaches can be used, depending on the final application. In this review, the role of microorganisms and biochar in bioremediation is thoroughly discussed. Along with its usage as an adsorbent for heavy metal ions and pesticides, biochar can also be utilized as an immobilization support for microorganisms. Carbonaceous materials have been frequently used as carrier materials for bacterial immobilization, to enhance the bioremediation efficiency of organic pollutants. Compared with expensive carbon materials, biochar is more competitive as a carrier material, as it is cheaper but has acceptably high porosity, which could provide shelter and nutrients for microbial cells, facilitating the colonization of microbial cells and the formation of microbial hot spots on the surface and in the pores of biochar. According to previous research, adsorption and entrapment are the most common methods for preparing the MCB. Toxic metal ions and pesticides have been successfully removed using immobilized cells. The key factors influencing the removal efficiencies are the pollutant’s concentration, incubation time, temperature, and pH.

The physical and chemical properties of biochar make it a suitable carrier/platform for microbial cell immobilization; however, this research area is still in its initial stages. The limitations related to the loss of activity of MCB and mass transfer potential have not been studied widely. Even though the immobilization of metal ions and pesticide-degrading microorganisms are cost-effective, stable, and environmentally friendly approaches, research can be conducted to enhance the treatment efficiency and improve the stability of microbial cells. The regeneration of the immobilized cells and recovery of the adsorbed pollutant can be improved. Most of the research focusing on immobilized microbes on biochar is mainly laboratory-based and involves the remediation of soil or an aqueous environment. The practical application of this in situ method is restricted, as the actual contamination sites are usually complicated. Research can be conducted to elucidate the heavy metal and pesticide degradation ability of a particular MCB from the soil and aqueous environment. The practical use of MCB can be further improved by increasing the efficacy and viability of the immobilized microbial cells and exploring approaches that would make the usage of MCB easier in contaminated sites. Moreover, the microbe-immobilized biochar can be employed in co-contaminated sites with heavy metals and pesticides for remediation. Genetically modified microorganisms are of increasing interest for the treatment of targeted pollutants. Therefore, further studies can be performed to genetically modify the microorganism for the targeted remediation of metal ions and pesticides, as well as to study the immobilization characteristics of these microbes on biochar.

## Figures and Tables

**Figure 1 molecules-28-00719-f001:**
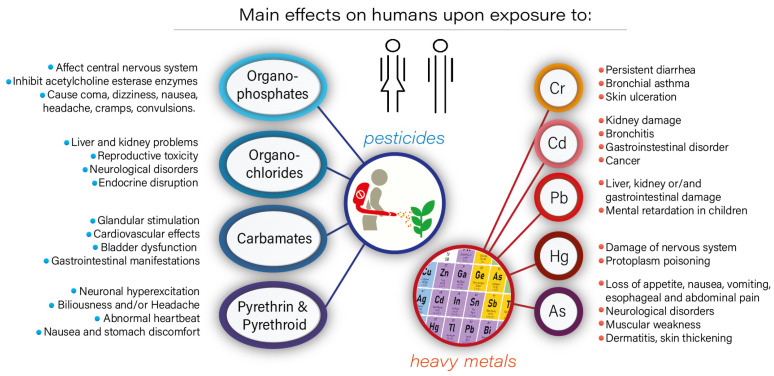
The effects of pesticides and heavy metal exposure on humans.

**Figure 2 molecules-28-00719-f002:**
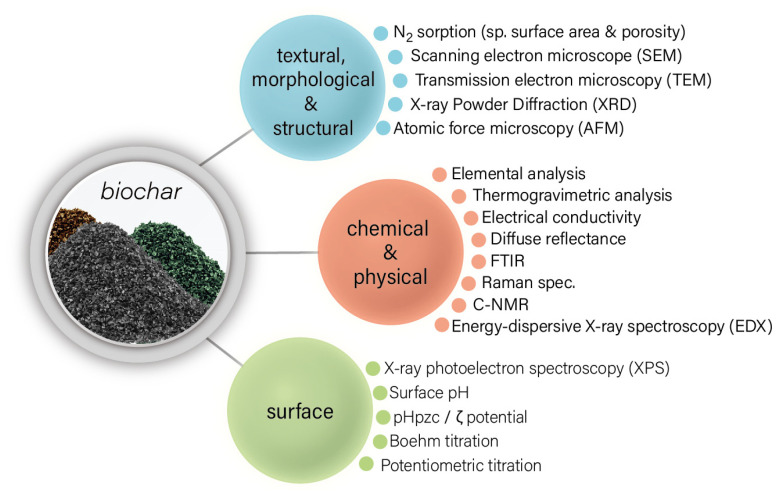
The predominant technique for the physicochemical characterization of biochar-based materials.

**Figure 3 molecules-28-00719-f003:**
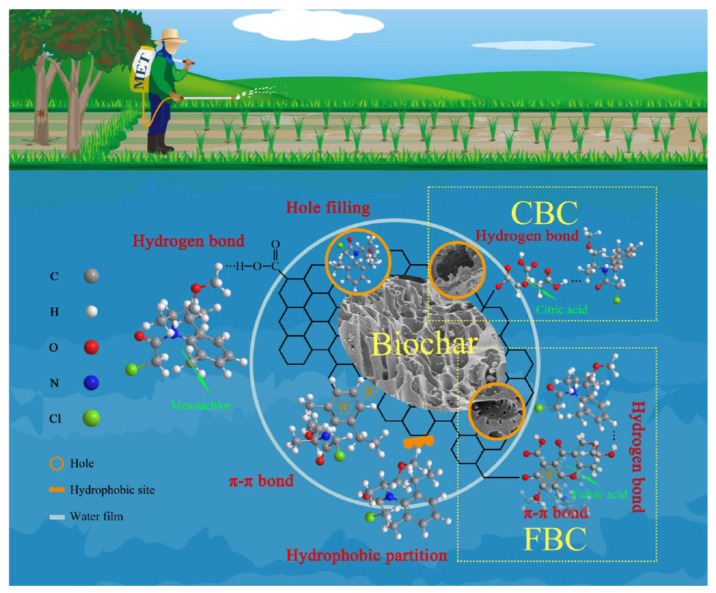
The fulvic acid- and citric acid-modified biochar adsorption mechanism for metolachlor in water Reprinted/adapted with permission from Ref. [54]. Copyright 2021, Elsevier.

**Figure 4 molecules-28-00719-f004:**
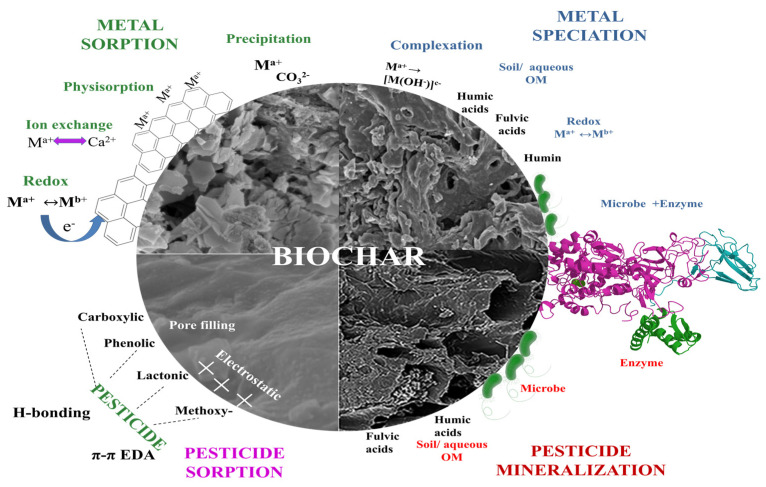
The overall involvement of biochar in heavy metal and pesticide remediation.

**Figure 5 molecules-28-00719-f005:**
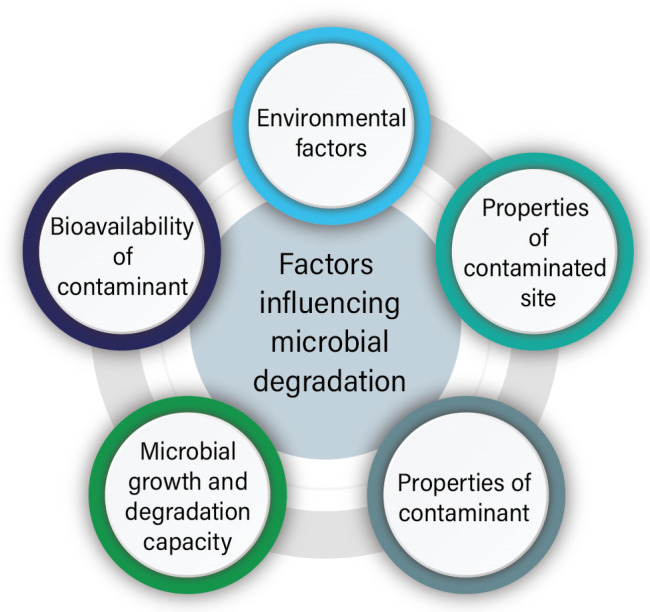
Factors influencing microbial degradation.

**Figure 6 molecules-28-00719-f006:**
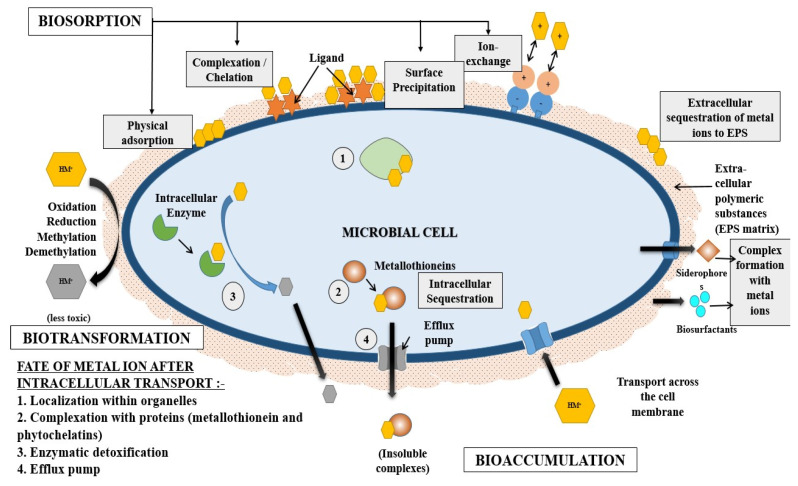
Mechanism of the microbial bioremediation of heavy metals.

**Figure 7 molecules-28-00719-f007:**
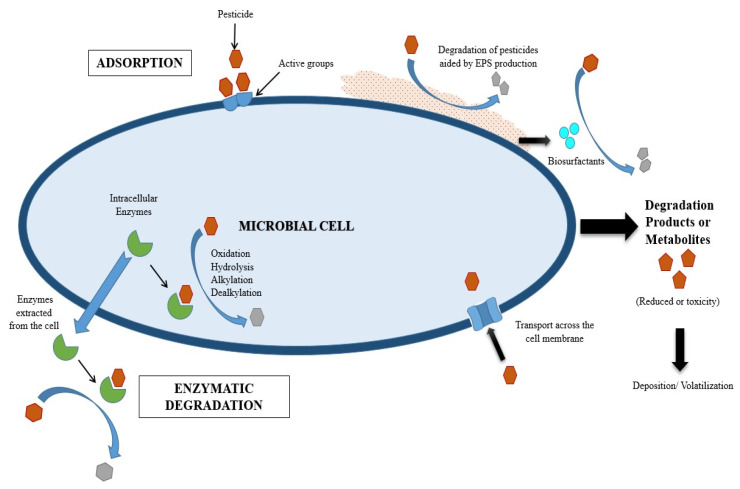
The mechanism of the microbial bioremediation of pesticide.

**Table 1 molecules-28-00719-t001:** Biochar types involved in heavy metal removal.

Biomass Type	Pyrolysis Temperature (°C)	Modification	Metal Ion	System	Adsorption Capacity (mg/g)	Reference
Crab shell	350	Fe-La doped	Sb^3+^	Water	498	[25]
Crab shell	350	Fe-La doped	SbO_6_^7−^	Water	337	[25]
Cattle manure	500	Fe-impregnated	Sb^5+^	Water	58.3	[37]
Wood chip	600	Sulfurized	Hg^2+^	Water	107.5	[37]
*Sesbania bispinosa*	450	MnO	AsO_4_^3−^	Water	7.35	[39]
*Sesbania bispinosa*	450	CuO	AsO_4_^3−^	Water	12.47	[39]
Rice straw	500	Thiol-modified	Cd^2+^	Soil	45.1	[41]
Rice straw	500	Thiol-modified	Pb^2+^	Soil	61.4	[41]
Lobster shell	600	HCl treatment	Cu^2+^	Water	71.4	[42]
Lobster shell	600	HCl treatment	Cd^2+^	Water	126	[42]
Peanut shell	600	MnO-embedded	Sb^3+^	Water	248	[44]
Corn straw	600	Fe-impregnated	HAsO_4_^2−^	Water	6.80	[45]
Cornstalk	550	Mg-Al-LDH	As^5+^	Soil	0.820	[46]
Cornstalk	550	Zn–Al-LDH	As^5+^	Soil	0.916	[46]
Cornstalk	550	Cu–Al-LDH	As^5+^	Soil	0.787	[46]
Canola straw	700	Steam activation	Pb^2+^	Water	195	[47]
Rice husk	500	HA/Fe-Mn oxide-loaded	Cd^2+^	Water	67.11	[48]
Rice husk	500	HA/Fe-Mn oxide-loaded	As^5+^	Water	35.59	[48]
Rice husk	1 kW (microwave)	Fe_3_O_4_-magnetic	Cr^6+^	Water	8.35	[49]
Pomelo peel	300	K_2_FeO_4_-promoted	Cr^6+^	Water	209.64	[50]
Sawdust	180	Amino-functionalized (HNO_3_, nicotinamide)	Sb^5+^	Water	241.92	[51]
Sawdust	180	Amino-functionalized (HNO_3_, nicotinamide)	Cr^6+^	Water	132.74	[51]

**Table 2 molecules-28-00719-t002:** Biochars utilized in pesticide removal.

Biomass Type	Pyrolysis Temperature (°C)	Modification	Pesticide	System	Adsorption Capacity (mg/g)	Reference
Cow manure	600	HCl/HF	Carbaryl	Water	~55	[24]
Dewatered sludge	700	-	Carbendazim	Soil	0.144	[26]
Leonardite	550	-	Alachlor	Water	3.802	[35]
Corn cob	600	HF	2,4-dichloro-phenoxyacetic acid	Water		[52]
Coconut fiber	600	HCl	Dichlorvos	Water	90.9	[53]
Walnut shell powder	700	Fulvic acid	Metolachlor	Water	99.01	[54]
Walnut shell powder	700	Citric acid	Metolachlor	Water	74.07	[54]
Bagasse	500	-	Carbofuran	Water	18.9	[55]
Switchgrass	425	Fe^3+^/Fe^2+^ magnetic	Metribuzin	Water	205	[56]
Switch grass	425	-	Metribuzin	Water	223	[56]

**Table 3 molecules-28-00719-t003:** Microorganisms that are involved in heavy metal removal.

Heavy Metal	Microorganism	Initial Heavy Metal Concentration	Incubation Time	Degradation Efficiency (%)	Reference
Bacteria					
Pb	*Bacillus cereus* BPS-9	-	48 h	77.57	[67]
	*Oceanobacillus profundus* KBZ 3-2	50 mg/L	24 h	97	[68]
	*Enterobacter* sp. FM-1	100 mg/L	24 h	93.85	[69]
Cr	*Bacillus subtilis SZMC 6179J*	55 mg/L	24 h	93.50	[70]
	*Pseudomonas aeruginosa*	20 ppm	21 days	89.67	[71]
	*Pseudomonas stutzeri* L1	100 mg/L	24 h	97	[72]
	*Bacillus cohnii*	100 mg/L	25 h	94	[73]
	*Bacillus licheniformis*	100 mg/L	25 h	95	[73]
Cd	*Weissella viridescens* ZY-6	NM	2 h	69.45–79.91	[74]
Zn	*Oceanobacillus profundus* KBZ 3-2	2 mg/L	24 h	54	[68]
Cu	*Pseudomonas aeruginosa*	15 ppm	14 days	90.89	[71]
As	*Bacillus* sp.	100 ppm	72 h	53.29	[75]
	*Aneurinibacillus aneurinilyticus*	100 ppm	72 h	50.37	[75]
Fungi					
Pb	*Trichoderma brevicompactum* QYCD-6	50 mg/L	5 days	97.5	[76]
Cr	*Trichoderma brevicompactum* QYCD-6	100 mg/L	5 days	31.83	[76]
Cd	*Penicillium notatum*	10 ppm	14 days	77.67	[71]
	*Trichoderma brevicompactum* QYCD-6	30 mg/L	5 days	20.13	[76]
Cu	*Trichoderma brevicompactum* QYCD-6	50 mg/L	5 days	64.46	[76]
Ni	*Aspergillus niger*	20 ppm	28 days	81.07	[71]
Microalgae					
Cd	*Desmodesmus* sp. MAS1	5 mg/L	7 days	>58%	[77]
	*Heterochlorella* sp. MAS3	5 mg/L	7 days	>58%	[77]
	*Chlorella vulgaris*	100 mg/L	5–15 min	Live cells—95.2 Dead cells—96.8	[78]
Zn	*Chlorophyceae* spp.	3 mg/L	3 h	91.9	[79]
Cu	*Chlorella vulgaris*	1.9–11.9 mg/L	12 days	39	[80]
	*Chlorophyceae* spp.	3 mg/L	10 min	88	[79]
As	*Scenedesmus almeriensis*	12 mg/L	3 h	40.7	[79]
Ni	*Chlorella vulgaris*	1.9–11.9 mg/L	12 days	32	[80]
Mn	*Scenedesmus almeriensis*	3 mg/L	3 h	99.4	[79]

**Table 4 molecules-28-00719-t004:** Microorganisms involved in pesticide removal.

Pesticide	Microorganism	Initial Pesticide Concentration	Incubation Time	Degradation Efficiency (%)	Reference
Bacteria					
Chlorpyrifos	*Pseudomonas nitroreducens* AR-3	100 mg/L	8 h	97	[98]
Chlorpyrifos	*Lactobacillus plantarum*	0.20–0.80 mg/kg	-	24.9–34.4	[99]
Malathion	*Escherichia coli* IES-02	50 ppm	4 h	99	[100]
Mesotrione	*Bacillus megaterium* Mes11	1 mM	5 h	99	[101]
Carbofuran	*Enterobacter* sp.	4 µg/ml	7 days	80	[102]
Fungi					
Chlorpyrifos	*Aspergillus sydowii* CBMAI 935	50 mg/L	30 days	32	[103]
Methyl parathion				80	
Profenos				52	
Pyrethroid mixture (cypermethrin, cyfluthrin, cyhalothrin)	*Aspergillus* sp.	500 mg/L	15 days	≈100	[104]
Microalgae					
Paraoxon, Malathion and Diazinon	*Coccomyxa subellipsoidea*	0.1 mg/ml	10 days	-	[105]
Atrazine	*Chlorella* sp.	40 µg/L	8 days	83.0	[106]
		80 µg/L		64.3	

**Table 5 molecules-28-00719-t005:** Microbial cell immobilized biochar for heavy metal and pesticide abatement.

Microorganism	Catalyst Support	Pollutant Type	Mechanism	System Water/Soil	Quantification of Heavy Metal Removal	Reference
*Bacillus* sp.TZ5	Coconut shell	Cd^2+^	Adsorption	Soil	48.49%	[141]
*Delftia* sp B9	Cornstalk	Cd^2+^	Adsorption	soil	0.33 mg/kg reduced to 0.06–0.13 mg/kg	[142]
*Chlorella* sp.	Water hyacinth	Cd^2+^	Adsorption	water	92.5%	[143]
*Leclercia adecarboxylata*	Rice hull	Pb^2+^	Entrapment	water	93%	[144]
*Bacillus subtilis*	Pig manure	Hg^2+^, Pb^2+^ co-contamination	Adsorption	water	69 mg/g Hg 112.3 mg/g Pb	[145]
*Bacillus subtilis*	Corn straw	Hg^2+^, Pb^2+^ co-contamination	Adsorption	water	53.7 mg/g Hg; 83.0 mg/g Pb	[145]
*Enterobacter* sp.	Rice husk BC	Pb^2+^	Adsorption	-	24.1%	[146]
*Enterobacter* sp.	Sludge BC	Pb^2+^	Adsorption	-	60.9%	[146]

## Data Availability

Not applicable.

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
