# Peer review of "Effective Usage of Biochar and Microorganisms for the Removal of Heavy Metal Ions and Pesticides"

_molecules, 2023, doi:10.3390/molecules28020719_

Round 1

Reviewer 1 Report

In the present manuscript, various bioremediation strategies utilizing biochar and microorganisms and how immobilized bacteria on biochar contributed to the improvement of bioremediation strategies were discussed. The full text has a relatively complete structure and content. However, there are still some problems about the contents. I recommended that this paper could be accepted for publication in your journal after major revision.

1. Line 69-83; Line 130-131: Following references with respect to biochar and its application in Fenton-like process may be considered by the authors: Water Research, 2019, 160, 238-248; Water Research, 2022, 221, 118797.

2. Line 133: The authors only described the pyrolysis production of biochar in detail, authors are suggested to add other synthesis methods of biochar and whether these different synthesis methods will affect the structure or properties of biochar.

3. Line 290-293: Authors are suggested to elaborate what are the possible roles of biochar as a carrier for adsorption? And give examples.

4. Line 316-317: Details should be given on how these mentioned factors affect microbial degradation.

5. Authors are suggested to change all tables to three-line tables.

6. What are the advantages of biochar over other carbon materials in the removal of heavy metals and pesticides or as a microorganisms carrier for the removal of heavy metals and pesticides?

Author Response

Reviewer #1

Question 1). Line 69-83; Line 130-131: Following references with respect to biochar and its application in Fenton-like process may be considered by the authors: Water Research, 2019, 160, 238-248; Water Research, 2022, 221, 118797.

The authors thank the Reviewer for the suggestion. The proposed references have been added at the revised version.

Li, L.; Lai, C.; Huang, F.; Cheng, M.; Zeng, G.; Huang, D.; Li, B.; Liu, S.; Zhang, M.M.; Qin, L.; et al. Degradation of naphthalene with magnetic bio-char activate hydrogen peroxide: Synergism of biochar and Fe–Mn binary oxides. Water Research 2019, 160, 238–248, doi:10.1016/j.watres.2019.05.081.

Liu, S.; Lai, C.; Zhou, X.; Zhang, C.; Chen, L.; Yan, H.; Qin, L.; Huang, D.; Ye, H.; Chen, W.; et al. Peroxydisulfate activation by sulfur-doped ordered mesoporous carbon: Insight into the intrinsic relationship between defects and 1O2 generation. Water Research 2022, 221, 118797, doi:10.1016/j.watres.2022.118797.

Question 2). Line 133: The authors only described the pyrolysis production of biochar in detail, authors are suggested to add other synthesis methods of biochar and whether these different synthesis methods will affect the structure or properties of biochar

Based on the useful comment, we have included various synthesis methods of biochar in section 2.1.1 and also elaborated on its effects on biochar structure and properties.

Biochar production usually involves biomass collected from various plant/animal sources or wastewater sludge and thermal treatment using oxygen-deficient conditions, particularly pyrolysis. Plant sources include olive pomace and rapeseed straw cereal waste, whereas animal sources include crustacean shells and animal manure [21–25]. Additionally, municipal wastewater sludge has also been used as biomass for biochar production [26]. The basic composition of biochar includes amorphous and crystalline carbon, containing graphene sheets and various aliphatic cyclic and aromatic groups as a matrix. Temperature and biomass source influence proportion and morphology [27], with fibrous biomass sources such as wheat/rice straw generating tubular structures. In contrast, the usage of sludge biochar prevented the formation of such structures in the biochar matrix [28].

Pyrolysis with oxygen-free conditions comprises the decomposition of lignocellulosic material, volatile matter release, and reducing carbonaceous material for plant biomass [29]. Types of pyrolysis conditions include slow, fast, microwave-assisted, hydro- and co-pyrolysis. Slow pyrolysis operates for hours at lower temperature conditions (300-700 °C), resulting in a higher output percentage of biochar content as com-pared to fast pyrolysis with lower residence time (<2-5 sec), higher temperature conditions, and lower output percentage of biochar. Increasing temperature can lead to higher carbon content, alkalinity, and specific surface area. In contrast, higher residence time can increase specific surface area due to prolonged temperature application.

Variations in high-temperature processes have also been tested for biochar pro-duction. Microwave-assisted pyrolysis for biochar generation has also been demonstrated, with variation in absorbable power observed for biochar property analysis, with demonstrated advantages of larger surface area and improved porosity characteristics [22] . Hydro pyrolysis usually is conducted in a temperature range of 250-550°C with hydrogen gas application, ensuring hydrocracking of biomass [19]. Co-pyrolysis involves multiple different biomass sources for biochar pyrolysis. The resultant physicochemical properties mainly depend on the biomass sources blending ratios and pyrolysis temperature, improving the biochar sample's pore structure [30]. Gasification is another method of generating biochar in the presence of oxygen/steam at 750-900 °C, with the products being syngas and a low biochar yield. Torrefaction is conducted with oxygen-deficient conditions similar to biochar. However, at temperatures of 200 -300 °C and residence time of fewer than 30 minutes. Another method explored extensively for biochar production is hydrothermal carbonization, with an operating temperature range from 160 to 800 °C (preferably lower temperatures) in the presence of water. The low-temperature environment results in higher O/C and H/C content along with functional groups on the biochar surface and lower aromaticity and porosity characteristics of such-generated biochar (hydrochar). The conversion of the non-carbonized (amorphous) part of biomass to carbonized form can be enhanced by increasing the pyrolysis temperature, which also increases the aromaticity, π electrons availability etc. [30]. Both the negative effect of pore size thermal shrinkage due to collapse of micropore walls and the positive effect of pore size increment due to removal of volatile matter can be observed with increasing temperature conditions. In-creasing pyrolysis temperatures also decrease the biochar’s stability for chemical oxidation resistance [31].

Question 3).  Line 290-293: Authors are suggested to elaborate what are the possible roles of biochar as a carrier for adsorption? And give examples.

We have modified the section 2.3 in accordance with the Reviewer’s suggestion.

       Biochar is often employed as a good carrier in improving the photocatalytic activity of metal oxides. As a stable and inexpensive carbonaceous material biochar effectively reduce the recombination rate of photogenerated electron-hole pairs due to its excellent conductive property. An et al. [61] developed biochar-supported α-Fe2O3/MgO composites for photocatalytic degradation of organophosphorus pesticides and obtained degradation efficiency of 90 % in 80 min. Huang et al. [62] utilized pristine and manganese ferrite modified biochar for Cu removal confirming the role of biochar being principally an oxide carrier instead of an adsorbent. In addition, preference of biochar as a carrier for photocatalysis and Fenton/ photo-Fenton processes has been prevalent due to its low-cost and high surface area characteristics. Utilization of lignin-biochar as catalyst support for LaFeO3 for catalytic photo-Fenton process had a positive effect on degradation efficiency of pollutants owing to enhanced adsorption capacity, reduction in charge transport resistance between LaFeO3 and lignin-biochar, and presence of oxygen-containing functional groups [63].

Question 4). Line 316-317: Details should be given on how these mentioned factors affect microbial degradation.

Based on the reviewer comment, we have explained about different factors affecting microbial degradation.

Bioremediation is possible only when microbial activity and growth are allowed by environmental conditions. In certain situations, environmental factors can be altered to allow microbial populations' growth to eliminate contaminants [11]. As shown in Figure 5, various factors influence microbial degradation, including

  • Environmental factors

      pH can affect bioremediation by changing metal bioavailability; for instance, a decrease in soil pH value generally causes an increase in metal bioavailability [66]. This is because, at lower pH, the exchangeable capacity between metal cations and H+ on the surface of soil particles is more prominent than at higher pH. Additionally, an optimum pH is essential for microbial growth, and some microbial degradation processes can be inhibited at extreme pH. Temperature is another crucial factor influencing the bioremediation of metals and pesticides [66]. The solubility of these contaminants is increased at higher temperatures, which leads to their increased bioavailability. The physical nature and chemical composition of several organic pollutants and its adsorption-desorption mechanism is governed by temperature. Temperature also influences microbial growth, activity, and degradation potential. Furthermore, the soil moisture content is another parameter which affects the bioremediation process. Low soil moisture content limits the growth and metabolism of microorganisms, while high values can reduce soil aeration.

(ii)  Type of microorganism and degradation capacity

    The microorganism selected for biodegradation should be able to survive in a high-contamination environment and be evaluated first for its degradation capacity before employing it for in-situ remediation. The survival of these strains can be ensured by providing favorable growth conditions. It is also important to note that microbial strains selected for pollutant removal may need to meet some ecological requirements. One such requirement is that the strains should be non-pathogenic. For instance, Staphylococcus aureus as a typical pathogen was resistant to many antibiotics and showed high bioremediation efficiency for heavy metals like Cr and U through bioprecipitation [67]. However, certain metabolites formed during the degradation of contaminants can be toxic. Therefore, deeper investigations of ecological security and metabolic functions of microbial cells are indispensable before their application in environmental pollution control.

(iii)  Bioavailability of the contaminants

     Bioavailability of the contaminants can be defined as the fraction of a contaminant in a specific environment that, either adsorbed or degraded by the microbial cells within a given time. The control of bioavailability is dependent on the diffusion, uptake, and desorption of the contaminants. Slow mass transfer of contaminants to degrading microbes reduces their bioavailability. The significance of bioavailability depends very much on the properties of the pollutant, microorganism, and characteristics of the contaminated site [11].

(iv)  Aerobic or anaerobic operating conditions 

      Depending on the type of organism and contaminant, bioremediation can be either aerobic or anaerobic. Most bioremediation systems work under aerobic conditions, but to effectively degrade recalcitrant molecules, it is better to run the microbial degradation under anaerobic conditions. Apart from the above-mentioned factors, the properties of the contaminated site (soil type, soil porosity, soil nutrients) and properties of contaminants (structure, hydrophobicity, recalcitrance, toxicity, solubility, and leaching ability) are also important in bioremediation.

Question 5). Authors are suggested to change all tables to three-line tables.

All the table were revised to three-line tables.

Question 6). What are the advantages of biochar over other carbon materials in the removal of heavy metals and pesticides or as a microorganism’s carrier for the removal of heavy metals and pesticides?

We have added the advantages of biochar over other carbon material in section 2.1 and section 5.

In section 2.1 Biochar production, properties, and characterization

Biochar has been well established as a low-cost adsorbent that has adsorption capacities similar to carbon-based adsorbents such as activated carbon, graphitic carbon nitride, graphene oxide etc., benefits being a) low-cost b) porous structure, c) shallow fabrication at large scale, d) eco-friendly nature, e) multiple functional groups (thus enabling both hydrophobic and polar interactions), f) ease in modifications etc. [19,20]. In addition, the preference for biochar as catalyst support for photocatalysis and Fenton/ photo-Fenton processes has been prevalent due to its low-cost and high surface area characteristics. In addition, the aromatic and other hetero-atom-containing functional groups present in biochar provide moieties capable of electron transfer and facilitate faster and more efficient degradation/ reduction of pollutants.

In section 5 Conclusion and future prospective

Carbonaceous materials have been frequently used as carrier materials for bacterial immobilization to enhance the bioremediation efficiency of organic pollutants. Compared with expensive carbon materials, biochar is more competitive to be a carrier material as it is cheaper with high porosity, which could provide shelter and nutrients for microbial cells, facilitating the colonization of microbial cells and the formation of microbial hot-spots on the surface and in pores of biochar.

Reviewer 2 Report

Journal: Molecules

Ms. ID.: molecules-2111905

Title: Effective usage of biochar and microorganism for removal of heavy metal ions and pesticides

Manikandan et al. compiled and discussed works focusing on the study of various bioremediation strategies utilizing biochar and microorganisms and how immobilized bacteria on biochar contributed to the improvement of bioremediation strategies. Besides, they provided a summary of the sources and harmful effects of pesticides and heavy metals. Based on the reviews described above, this study aims to outline the future scopes of this field. It is a very interesting manuscript. It fits well with the scope of the Journal. I consider the manuscript suitable for publication, but I also believe some important improvements are needed. The list of specific issues that should be addressed is listed below.

-To call bioremediation environmentally friendly is not the appropriate way. There are many potentially harmful effects of the introduction of new species in the environmental niches. This should be elaborated better. Microorganisms could produce nus-products more toxic than the parent compound. This issue has to be addressed properly.

-Figure 1. The effects of pesticides and heavy metals on human organisms should be provided with better care and precision. There are many more. The effects presented in the figure are not the only ones and not the most notorious.  

-Figures 3, 4 and 5 are blurry.

-The authors should elaborate in more detail on the benefits of using the microorganisms immobilized onto biochar, and not each component individually. I find it not obvious from the present manuscript.

Author Response

Please see the attached file with our detailed reply to all comments/suggestions/corrections. 

Round 2

Reviewer 1 Report

Thanks for the response from authors! Authors revised the manuscript in detail according to the comments of reviewers, and the quality of this paper was greatly improved. I recommended that this paper could be accepted for publication in this journal.

Reviewer 2 Report

The authors addressed all my comments. I recommend the manuscript for publication in present form.